# Gender Differences in the Impact of Cognitive Function on Health Literacy among Older Adults with Heart Failure

**DOI:** 10.3390/ijerph15122711

**Published:** 2018-12-01

**Authors:** Jong Kyung Lee, Youn-Jung Son

**Affiliations:** 1College of Nursing, Dankook University, Cheonan 31116, Korea; kyunglee@dankook.ac.kr; 2Red Cross College of Nursing, Chung-Ang University, Seoul 06974, Korea

**Keywords:** heart failure, aged, gender, cognition, health literacy

## Abstract

Heart failure (HF)-related cognitive decline is a common condition and may be associated with health literacy. However, gender differences in this context have not been explored fully. This secondary data analysis aimed to identify gender differences in the impact of cognitive function on health literacy among older patients with HF. A total of 135 patients (75 men and 60 women) with a mean age of 73.01 ± 6.45 years were recruited. Older women with HF had higher cognitive impairment (15%) and inadequate health literacy (56.7%) compared to men. Cognitive function was the strongest predictor of health literacy in men (*β* = 3.668, *p* < 0.001) and women (*β* = 2.926, *p* = 0.004). Notably elderly women are likely to face double the burden of the influence of cognitive function on health literacy in comparison with men. It is necessary to assess cognitive function and health literacy during HF illness trajectories on a regular basis. Healthcare professionals working with patients with HF should be aware of gender differences in cognitive function and health literacy and the importance of assessing these factors.

## 1. Introduction

Heart failure (HF), one of the most common chronic diseases worldwide, is a deliberating and progressive condition [1,2]. HF predominantly affects older adults with multiple health problems [3,4], and can lead to many adverse health outcomes including poor quality of life, frequent hospitalization, and high mortality [5]. Thus, effective self-care is critical in improving patient health outcomes [6,7]. However, despite its importance, most HF patients have trouble taking part in the necessary self-care activities that are recommended in clinical guidelines [8,9,10]. Moreover, self-care behaviors among older patients with HF are more complicated owing to co-morbidities and age-related problems such as hearing, visual, and cognitive impairments.

Cognitive impairment means that a person has trouble remembering recent events, focusing or making decisions that affect their activities of daily living [7,8]. Importantly, cognitive impairment has been identified as a common consequence of HF [8,10]. HF-related cognitive dysfunction is common in 73–80% of HF patients [8,11]. Cognitive impairment may interfere with the ability and awareness to recognize worsening symptoms, to perform self-care activities adequately, and to seek individualized treatment [9,12]. Previous studies suggest that healthcare professionals should consider the impact of cognitive impairment on self-care behaviors and utilizing healthcare services among older HF patients [3,8,12]. Some studies on gender differences in cognitive function reported that women score better than men on tests of verbal abilities, and in contrast, men perform better than women on tests of visuospatial skills in general [8,10]. According to a recent study [13], women showed patterns with slower cognitive decline and poorer executive function than men. A retrospective study on patients awaiting cardiac transplant have shown that female patients had significantly greater cognitive impairments in psychomotor speed and language than male patients [14]. However, there are few empirical studies that have examined gender differences in cognitive impairments in HF [15]. Only one cross-sectional study has directly examined gender differences in cognitive function among HF patients [16]. This study found that men and women with HF did not differ in attention and executive function [16].

Health literacy means the ability of an individual to access health care services and use health information to make appropriate decisions [17]. HF patients with inadequate health literacy may experience trouble accessing and utilizing health-related information [18,19]. Previous studies have shown that inadequate health literacy was associated with limited knowledge, non-adherence to treatment, and adverse health outcomes [2,18]. Specific domains of cognitive functioning that might play a role in the health literacy of older adults are memory, information processing speed, and mental flexibility [20,21]. Thus, cognitive impairments in memory may impede a patient’s ability to remember basic information about self-care activities in HF patients as well as general populations [20,21].

To date, there are a number of studies on cognitive function and health literacy among the general population [20,22]. However, relatively less is known about possible gender differences in the role of cognitive function on health literacy as a key indicator of self-care in older patients with HF. Therefore, the aim of this study is to identify the gender differences in the impact of cognitive function on health literacy among older adults with HF.

## 2. Materials and Methods

### 2.1. Study Design and Participants

This was a secondary analysis of cross-sectional data from the study investigating risk factors including symptom clusters for re-hospitalization among Korean HF outpatients [23]. A total of 135 patients (75 men and 60 women) who provided data on the variables of interest in this study such as cognitive function and health literacy were included in this analysis. HF was diagnosed by a cardiologist, an expert in HF based on clinical history and physical examination results, and supplemented by transthoracic echocardiography findings. The eligibility criteria for the participants included the diagnosis of HF at least six months prior to the study, aged 65 or above, alertness and willingness to participate, residence at home and caring for self, and understanding spoken Korean. Patients with a life expectancy of less than six months or an active listing for heart transplantation, established neurodegenerative disorder such as Alzheimer’s or Parkinson’s diseases, positive history suggestive of a stroke, and established clinical diagnosis of depression or other psychiatric ailments were all excluded from the study. The sample size was in accordance with the G*Power 3.1.9.2 calculation for a difference between the two independent groups (total sample size 128, effect size = 0.5 (medium), significance value = 0.05, 1 − *β* = 0.8) in the two-tailed method. Therefore, the sample size in this study was appropriate.

### 2.2. Ethical Consideration and Data Collection

All procedures were performed in accordance with the Declaration of Helsinki. Ethical approval for secondary analysis was granted by the Institutional Review Board (201809-SB-070). Informed consent was formally obtained from each participant before data collection. Patients completed a structured one-on-one interview with two trained research assistants. A research assistant approached the participants in the waiting area of the cardiac outpatient center. The research assistants provided information on the purpose of the study and informed patients that their responses would be maintained as anonymous and that they could withdraw at any time.

### 2.3. Instruments

#### 2.3.1. Patient Characteristics

Data collected on patients’ socio-demographic characteristics included gender, age, spouse, education, and monthly income. This information was collected through face-to-face interviews. Disease-related characteristics were collected through a review of electronic medical records. This data included the duration of diagnosed HF, New York Heart Association (NYHA) classification, left ventricular ejection fraction (LVEF, %), physician-diagnosed co-morbidities (hypertension, diabetes, coronary artery disease (CAD)), and prescribed cardiovascular medication.

#### 2.3.2. Cognitive Function

Cognitive function was measured using the Modified Mini-Mental Status Examination (3MS) [24]. The Korean version of the 3MS was used in this study [25]. The 3MS is a well-established and brief cognitive test including attention, orientation to time, memory, calculation, and language when compared to the Mini-Mental State Exam (MMSE) [24]. Moreover, the 3MS has been useful and sensitive in measuring cognitive function among HF patients [26]. The 3MS score ranges from 0 to 100 points. Higher scores indicate better cognitive function [24]. Cognitive impairment was defined as having a 3MS score < 80 [27].

#### 2.3.3. Health Literacy

To assess health literacy, we adopted a single-item health literacy (SHL) scale. The screening question was, “How confident are you about filling out medical forms by yourself?” This brief question was developed by Chew et al. [28] for screening patients’ self-reported difficulties in evaluating health-related performance. Responses were rated on a five-point scale: (1) = not at all, (2) = a little bit, (3) = somewhat, (4) = quite a bit, or (5) = extremely. Higher scores indicated higher levels of health literacy. A score below 3 represented an inadequate level of health literacy [29]. A single-item health literacy as a quick assessment, was validated in older adults [30].

### 2.4. Data Analyses

Data were analyzed using IBM SPSS for windows, version 23.0 (IBM, Armonk, NY, USA). Descriptive statistics of mean, standard deviation, frequency, and percentage were used to summarize data on patient characteristics. Independent t-test and chi-square tests were performed to assess the gender differences in the cognitive function and health literacy levels among participants. Hierarchical linear regression analysis was performed to investigate the impact of cognitive function on health literacy in men and women. All the socio-demographic and clinical variables that were statistically significant in Table 1 were entered in the first step. In the second step, cognitive function was added. Dummy variables were created for the independent variables with nominal or ordinal levels of measurement. Tolerance and variance inflation factor (VIF) were used to check for multicollinearity. A VIF value less than 10 and a tolerance value greater than 0.2 were acceptable. All tests were performed at a statistical significance level of *p* = 0.05.

## 3. Results

### 3.1. Patient Characteristics

Patient characteristics are listed in Table 1. Among the 135 patients with HF, 60 (44.4%) were women. The women were around 3.9 year older than men (*p* = 0.001). Compared with men, women were more likely to have no spouse (*p* < 0.001), be less educated (*p* = 0.006), have low income (*p* = 0.006), and anemia (*p* = 0.011). In contrast, men were more likely to have a lower proportion of LVEF (%) < 40 (*p* = 0.034), and higher proportion of CAD (*p* = 0.001). However, there were no statistically significant differences in the duration of HF diagnosis, NYHA functional class, having hypertension and diabetes, and taking prescribed medication.

### 3.2. Differences in Cognitive Function and Health Literacy by Gender

The mean score of cognitive function and health literacy of the total sample were 87.60 (SD = 10.32) and 3.59 (SD = 1.18) respectively (Table 2). In our sample, the overall prevalence of cognitive impairment and inadequate health literacy were 8.1% and 45.9% respectively.

Men had a higher cognitive function (89.92 ± 8.08) than women (84.70 ± 12.02). Particularly, the proportion of cognitive impairment in men (2.7%) was relatively lower than in women (15.0%), which was statistically significant (*p* = 0.009). Of the total, 47 men (62.7%) had marginal or adequate levels of health literacy, which was higher than among women (43.3%).

### 3.3. Impact of Cognitive Function and Health Literacy in Men and Women

Hierarchical linear regression results are presented in Table 3. For men, age (*β* = −0.239, *p* = 0.040) and cognitive function (*β* = 3.668, *p* < 0.001) were significantly associated with health literacy. In step 1, the model explained 42.1% of the variance in health literacy. Upon adding cognitive function in step 2, the overall model explained 55.6% of the variance in health literacy (F = 11.298, *p* < 0.001). Thus, cognitive function independently explained 13.5% of the variance in health literacy among men.

For women, cognitive function (*β* = 2.926, *p* = 0.004) was the strongest predictor of health literacy. In step 1, the model explained 22.8% of the variance in health literacy. Upon adding cognitive function in step 2, the overall model explained 54.8% of the variance in health literacy (F = 8.954, *p* < 0.001). Thus, cognitive function independently explained 29.7% of the variance in health literacy among women.

## 4. Discussion

In our cross-sectional study, the global measure of cognitive function (3MS) indicated that the mean scores in men and women were 89.92 and 84.70 respectively, presenting 2.7% and 15.0% prevalence of cognitive impairments scoring under 80, respectively. The prevalence of cognitive impairment within our sample was similar to the findings of previous studies investigating older outpatients with HF using the 3MS [30,31]. In contrast, this was a lower prevalence than in previous studies which reported 34.5% to 40% of participants had impaired cognitive function on the 3MS [20,27]. This discrepancy might be the reason why most of our sample was a relatively better functional class with NYHA I and II (86.7%) and short HF diagnosis duration with an average of 5.04 years. According to prior studies, older adults who were diagnosed with HF experienced, on average, more rapid subsequent decline in cognitive performance than people who had no history of HF [11]. Furthermore, the incidence of cognitive impairments among HF patients might vary widely depending on the population studied, the type or severity of the disease itself and the study design used [32,33]. Further investigation is particularly necessary to identify which cognitive assessment tool is superior in studying older people with HF.

Elderly women in our sample had lower levels of cognitive function with higher prevalence of cognitive impairment than men. Our result confirmed that women with HF had lower cognitive function when compared to men [13,34]. However, it was not in concordance with previous studies which showed that women had an advantage over men in verbal memory tasks [35]. There was no difference in the prevalence of cognitive impairment between men and women with HF [16]. Possible reasons for this might be genetic factors, socio-economic status, lifestyle, and hormonal, psychological, and neurobiological factors [8]. Elderly women in the study had lower educational attainment and monthly income in comparison with men. Lower educational attainment and personal income may influence increased risk of cognitive impairment because lower socio-economic status is associated with poor nutritional intake, less access to healthcare services, less social activity and less interpersonal communication, all of which can lead to social isolation [18,36]. On the other hand, other studies have noted that men may experience faster verbal memory decline with the earlier onset of CAD and a higher risk for incident mild cognitive impairment [13]. However, even though there was a higher prevalence of CAD among men in this study, cognitive impairment was significantly prevalent among women. Accordingly, further research is necessary to identify the factors that influence cognitive function in men and women.

Similar to Cajita et al. [18], we found that inadequate health literacy is high among older adults with HF. About 45.9% of the participants had inadequate levels of health literacy. Moreover, supporting the findings in Fabbri et al. [2], this study found that inadequate health literacy was more prevalent among women (56.7%) than among men (37.3%). However, the findings of this study were not consistent with Peterson et al. [37] and Wu et al. [38] who found that there was no significant association between gender and health literacy. This inconsistency might be a result of different cultural backgrounds involved in the study. In traditional Korean society, there has been a huge educational gap between men and women. Furthermore, there is also a large generation gap. Educational attainment has been limited among older women in South Korea for several decades, which means that women were confined to their home [39]. This result indicates that older Korean women with less than middle school education and low income might have a greater need for improvement in health literacy.

Our main findings showed that cognitive function remained the strongest predictor of health literacy in both men and women even after controlling for socio-demographic and disease-related factors. This result was consistent with Hawkins et al. [20] and Morrow et al. [40], which examined the relationship between cognitive function and health literacy among HF patients. In particular, women are likely to face double the burden of the influence of cognitive function on health literacy in comparison with men. Poor opportunities for education might be closely linked to low cognitive function and health literacy among elderly women. A stronger relationship is detected between cognitive function and health literacy in women when compared to men. This suggests that the cognitive status of women patients should be prioritized in order for interventions for health literacy to be maximally effective.

The presence of anemia was significantly associated with inadequate levels of health literacy among men and women, even though there was no significant association in the hierarchical linear regression. Jaarsma et al. [5] and Riegel et al. [6] reported that anemia is a predictor of mortality for patients with HF regardless of whether they have a reduced LVEF. Little is known about the association between gender and anemia and health outcomes in patients with HF. Previous study showed that the presence of anemia has been considered a risk factor for poor mobility, increased frailty, and decreased executive function among women [41]. More women in our study had under 40% LVEF and anemia in comparison with men. This finding indicates the possibility that lower cardiac contractility and the presence of anemia can increase the risk of cognitive impairment and inadequate health literacy. It can lead to a tremendous burden for elderly women with HF. For this reason, healthcare professionals should be aware of this issue to ensure that self-management tasks for both older men and women with HF have appropriate cognitive and literacy factors.

## 5. Limitations

There are several limitations in this study. First, the cross-sectional design used a convenience HF sample from a single cardiac center. This precludes the inference of causal relationships. Second, most participants in the study were clinically stable outpatients with chronic HF based on NYHA functional class. This limits the generalizability of the results. Third, this study used a single assessment tool. Future studies should use a battery of cognitive tests rather than a single assessment tool to identify the role of specific cognitive domains such as attention and working memory for cognitive impairment. Finally, this study examined the performance of a SHL screening question alone. Although the SHL screening item is helpful for the quick assessment of the health literacy levels among HF patients, the use of a single screening question is not enough.

## 6. Conclusions

We found that cognitive function was the strongest predictor of health literacy in both men and women even after controlling for socio-demographic and disease-related factors. Most importantly, this study highlights that the decline in cognitive function among elderly women with HF might have a more significant impact on health literacy levels when compared to the situation among men. Findings from this suggest that healthcare professionals working with patients with HF need to be aware of the gender differences, particularly in the level of cognitive function and health literacy among older patients with HF. It is necessary to assess cognitive function and health literacy during HF illness trajectories on a regular basis. In addition, effective interventions based on modifiable factors such as knowledge level and co-morbidities including anemia in both men and women can contribute toward the improvement of cognitive function and health literacy among older patients with HF. Longitudinal investigation is necessary to investigate gender differences in the relationships between cognitive function and health literacy with a larger sample, with due consideration for diverse socio-demographic variables and cultural differences.

## Figures and Tables

**Table 1 ijerph-15-02711-t001:** Gender differences in socio-demographic and clinical characteristics.

Characteristics	Total (*n* = 135)	Men (*n* = 75)	Women (*n* = 60)	t or χ^2^ (*p*)
	Mean ± SD or *n* (%)	Mean ± SD or *n* (%)	Mean ± SD or *n* (%)	
Age (years)	73.01 ± 6.45	71.29 ± 6.45	75.15 ± 5.99	−3.56 (0.001)
Spouse, yes	101 (74.8)	66 (88.0)	35 (58.3)	15.57 (<0.001)
Education				
Below elementary school	74 (54.8)	32 (42.6)	42 (70.0)	10.31 (0.006)
Above Middle school	61 (45.2)	43 (57.4)	18 (30.0)	
Monthly income (KRW)				
<1,000,000	92 (75.6)	50 (66.7)	42 (70.0)	10.31 (.006)
≥1,000,000	43 (24.4)	25 (33.3)	18 (30.0)	
Duration of HF diagnosis (years)	5.04 ± 57.89	3.57 ± 9.01	3.48 ± 9.60	−0.53 (0.594)
NYHA class				
I	39 (28.9)	21 (28.0)	18 (30.0)	2.77 (0.250)
II	78 (57.8)	47 (62.7)	31 (51.7)	
III–IV	18 (13.3)	7 (9.3)	11 (18.3)	
LVEF(%), <40	54 (40.0)	24 (32.0)	30 (50.0)	4.50 (0.034)
HTN, yes	67 (49.6)	37 (49.3)	30 (50.0)	0.01 (0.939)
DM, yes	37 (27.4)	22 (29.3)	15 (25.0)	0.32 (0.575)
CAD, yes	102 (75.6)	65 (86.7)	37 (61.7)	11.28 (0.001)
Anemia, yes	35 (25.9)	13 (17.3)	22 (36.7)	6.49 (0.011)
ACEI, yes	30 (22.2)	16 (21.3)	14 (23.3)	0.08 (0.781)
ARB, yes	23 (17.0)	9 (12.0)	14 (23.3)	3.03 (0.082)
Diuretic, yes	55 (40.7)	27 (36.0)	28 (46.7)	1.57 (0.210)
Beta blocker, yes	86 (63.7)	52 (69.3)	34 (56.7)	2.31 (0.128)

KRW = Korean Won; HF = heart failure; NYHA = New York Heart Association; LVEF = left ventricular ejection fraction; HTN = Hypertension; DM = diabetes mellitus; CAD = coronary artery disease; ACEI: angiotensin converting enzyme inhibitors.

**Table 2 ijerph-15-02711-t002:** Difference in cognitive function and health literacy by gender.

Variables (Range)	Total (*n* = 135)	Men (*n* = 75)	Women (*n* = 60)	t or χ^2^ (*p*)
	Mean ± SD or *n* (%)	Mean ± SD or *n* (%)	Mean ± SD or *n* (%)	
Global cognitive Function (1~100)	87.60 ± 10.32	89.92 ± 8.08	84.70 ± 12.02	2.88 (0.005)
Normal (>80)	124 (91.9)	73 (97.3)	51 (85.0)	6.78 (0.009)
Impairment (≤80)	11 (8.1)	2 (2.7)	9 (15.0)	
Health literacy (1–5)	3.59 ± 1.18	3.84 ± 1.07	3.28 ± 1.25	2.79 (0.006)
Inadequate (≤3)	62 (45.9)	28 (37.3)	34 (56.7)	6.49 (0.011)
Marginal/Adequate (4–5)	73 (54.1)	47 (62.7)	26 (43.3)	

**Table 3 ijerph-15-02711-t003:** Hierarchical linear regression analysis for health literacy in heart failure patients according to gender.

Gender	*Predictors*	Step 1			Step 2		
		*β*	t (*p*)	95% CI	*β*	t (*p*)	95% CI
Men (*n* = 75)	Age (years)	−0.429	−3.535 (0.001)	−1.503 to −0.418	−0.239	−2.100 (0.040)	−1.042 to −0.026
	Spouse (1 = yes)	−0.107	−1.149 (0.255)	−2.294 to −0.617	−0.140	−1.699 (0.094)	−2.390 to 0.193
	Education (1 = above middle school)	0.304	2.902 (0.005)	0.489 to 2.644	0.169	1.694 (0.095)	−0.156 to 1.900
	Monthly income (1 = ≥1,000,000 KRW)	0.044	0.441 (0.661)	−0.837 to 1.312	0.035	0.398 (0.692)	−0.755 to 1.131
	Ejection fraction (1 = ≥40)	−0.102	−1.116 (0.268)	−0.081 to 0.023	−0.078	−0.958 (0.342)	−0.068 to 0.024
	Coronary artery disease (1 = yes)	0.070	0.753 (0.454)	−0.864 to 1.910	0.057	0.698 (0.488)	−0.792 to 1.644
	Anemia (1 = yes)	−0.263	−2.327 (0.023)	−3.280 to −0.253	−0.111	−1.307 (0.195)	−1.890 to 0.393
	Global cognitive function				0.515	3.668 (<0.001)	0.074 to 0.252
		Adjusted R^2^ = 0.421, F (*p*) = 8.673 (<0.001)			Adjusted R^2^ = 0.556, R^2^ change = 0.135, F (*p*) = 11.298 (<0.001)		
Women (*n* = 60)	Age (years)	0.058	0.386 (0.701)	−0.581 to 0.858	0.028	0.238 (0.813)	−0.492 to 0.624
	Spouse (1 = yes)	0.003	0.021 (0.983)	−1.572 to 1.605	−0.002	−0.017 (0.987)	−1.250 to 1.229
	Education (1 = above middle school)	0.387	2.696 (0.009)	0.599 to 4.083	0.189	1.655 (0.104)	−0.245 to 2.534
	Monthly income (1 = ≥1,000,000 KRW)	0.102	0.748 (0.458)	−1.395 to 3.053	0.160	1.533 (0.132)	−0.405 to 3.017
	Ejection fraction (1 = ≥40)	−0.043	−0.367 (0.715)	−0.082 to 0.056	0.012	0.131 (0.897)	−0.050 to 0.057
	Coronary artery disease (1 = yes)	0.062	0.512 (0.611)	−1.037 to 1.747	0.119	1.266 (0.211)	−0.397 to 1.749
	Anemia (1 = yes)	−0.355	−2.887 (0.005)	−3.451 to −0.625	−0.149	−1.412 (0.164)	−2.067 to 0.358
	Global cognitive function				0.553	2.926 (0.004)	0.130 to 0.701
		Adjusted R^2^ = 0.228, F (*p*) = 3.494 (0.004)			Adjusted R^2^ = 0.548, R^2^ change = 0.297, F (*p*) = 8.954 (<0.001)		

KRW = Korean Won; CI = confidence interval.

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
