# Peer review of "Gender Differences in the Impact of Cognitive Function on Health Literacy among Older Adults with Heart Failure"

_ijerph, 2018, doi:10.3390/ijerph15122711_

Round 1
Reviewer 1 Report
The topic is relevant and the manuscript is well written. The literature review provides justification for why the study was conducted. The methods are adequately described, however reliability and validity of instruments need to be reported. As mentioned in the limitation section, I have concern for the HL screening tool used for this study, given that 70% women had education levels below elementary school. Authors should mention why they chose this tool over other HL measurement instruments. The results were presented clearly and discussion compared results to existing literature.
Author Response
Response to reviewer 1 comments
Manuscript ID ijerph-401303: Gender Differences in the Impact of Cognitive Function on Health Literacy among Older Adults with Heart Failure
We appreciate the very helpful comments on the initial draft. The manuscript has been strengthened significantly with your guidance. We have tried our best to make revisions as reviewer comments have mentioned. Please positively consider above revisions according to your comments.
---------------------------------------------------------------------------------------------------------------------------------
The topic is relevant and the manuscript is well written. The literature review provides justification for why the study was conducted. The methods are adequately described, [Comment 1] however reliability and validity of instruments need to be reported. As mentioned in the limitation section, I have concern for the HL screening tool used for this study, given that 70% women had education levels below elementary school. Authors should mention why they chose this tool over other HL measurement instruments. The results were presented clearly and discussion compared results to existing literature.
Authors’ response: Thank you for your helpful comment. A single-item health literacy has been reported that it was valid and reliable in previous studies [28,29]. We added new reference [30] and a sentence as follows:
On line 115-116, page 3- A single-item health literacy as a quick assessment was validated in older adults [30]
[30] Woods, N. K.; Chesser, A. K. Validation of a single health literacy screening tool for older adults. Gerontology and Geriatric Medicine. 2017, 3, 1-4, doi:10.1177/2333721417713095
Thank you for your meaningful comments once again.

Reviewer 2 Report
The paper entitled "Gender Differences in the Impact of Cognitive Function on Health Literacy Among Older Adults with Heart Failure" evaluated gender differences in sociodemographic, cognitive function and health literacy in 135 patients. This research entails a significant advance in the comprehension of gender differences regarding the association between heart failure, cognitive function and health literacy. Some minor issues should be adressed by authors prior to the paper will be published.
Abstract
There is a mistake in the line 11 (heath).
Introduction
The definition of cognitive impairment should be included in the Introduction section.
In lines 42-43 authors said: "However, only a few studies have directly examined possible gender differences in cognitive function among HF patients." What type of studies? These studies have been described previously? If not, authors should explain in detail the previous research regarding gender differences in cognitive functioning among HF patients.
The relationship between health literacy and cognitive functioning in HF patients should be described deeply.
Results
In Table 1, the variable "spouse" seems not to be appropriate for the female group.
Authors should replicate the obtained significant differences in Cognitive Function and Health Literacy controlling for possible confounders, mainly those sociodemographic variables in which it has been found differences between groups (age, marital status, education, income and disease related factors).
Author Response
Response to reviewer 2 comments
Manuscript ID ijerph-401303: Gender Differences in the Impact of Cognitive Function on Health Literacy among Older Adults with Heart Failure
We appreciate the very helpful comments on the initial draft. The manuscript has been strengthened significantly with your guidance. We have tried our best to make revisions as reviewer comments have mentioned. Please positively consider above revisions according to your comments.
-----------------------------------------------------------------------------------------------------------------
The paper entitled "Gender Differences in the Impact of Cognitive Function on Health Literacy among Older Adults with Heart Failure" evaluated gender differences in sociodemographic, cognitive function and health literacy in 135 patients. This research entails a significant advance in the comprehension of gender differences regarding the association between heart failure, cognitive function and health literacy. Some minor issues should be addressed by authors prior to the paper will be published.
Thank you for your encouraging comments. We have tried to make our best efforts to revise some parts according to your comments.
Comment 1. Abstract: There is a mistake in the line 11 (heath).
=> Authors’ response: We corrected misspelling : heath=> health
Comment 2. Introduction
2.1 The definition of cognitive impairment should be included in the Introduction section.
=> Authors’ response: According to your comments, we added the definition of cognitive impairments on line 33-34, page 1 as follows.
Cognitive impairment means that a person has trouble remembering recent events, focusing or making decisions that affect their activities of daily living
2.2 In lines 42-43 authors said: "However, only a few studies have directly examined possible gender differences in cognitive function among HF patients." What type of studies? These studies have been described previously? If not, authors should explain in detail the previous research regarding gender differences in cognitive functioning among HF patients.
=> Authors’ response: Thank you for your valuable comments. According to your comment, we added in a more detail as follows.
On line 44-49, page 2: A retrospective study on patients awaiting cardiac transplant have shown that female patients had significantly greater cognitive impairments in psychomotor speed and language than male patients [14]. However, there are few empirical studies that have examined gender differences in cognitive impairments in HF [15]. Only one cross-sectional study have directly examined gender differences in cognitive function among HF patients [16]. This study have found that men and women with HF did not differ no attention and executive function [16].
2.3. The relationship between health literacy and cognitive functioning in HF patients should be described deeply.
=> Authors’ response: According to your comments, we added regarding the relationship between cognitive function and health literacy as follows.
On line 54-58, page 2: Specific domains of cognitive functioning that might play a role in the health literacy of older adults are memory, information processing speed, and mental flexibility [20,21]. Thus, cognitive impairments in memory may impede a patient’s ability to remember basic information about self-care activities in HF patients as well as general populations [20, 21].
Comment 3. Results
1) In Table 1, the variable "spouse" seems not to be appropriate for the female group.
=> Authors’ response: In this study, around 58.3% of women had husband. So this value or the meaning of spouse would be appropriate.
2) Authors should replicate the obtained significant differences in Cognitive Function and Health Literacy controlling for possible confounders, mainly those sociodemographic variables in which it has been found differences between groups (age, marital status, education, income and disease related factors).
=> Authors’ response: According to your comments, we emphasized the significant impact of cognitive impairments on health literacy in HF patients in the first paragraph of Conclusion as follows.
On line 252-253, page 8: We found that cognitive function was the strongest predictor of health literacy in both men and women even after controlling for socio-demographic and disease-related factors. Most importantly, this study highlights that the decline in cognitive function among elderly women with HF might have a more significant impact on health literacy levels when compared to the situation among men.
Thank you for your meaningful comments once again.
